# Modeling Legal Reasoning:
# LM Annotation at the Edge of Human Agreement

**Rosamond Thalken** [1]    **Edward H. Stiglitz** [2]    **David Mimno** [1]    **Matthew Wilkens** [1]

[1] Cornell University    [2] Cornell Law School

{ret85, js2758, mimno, wilkens}@cornell.edu

## Abstract

Generative language models (LMs) are increasingly used for document class-prediction tasks and promise enormous improvements in cost and efficiency. Existing research often examines simple classification tasks, but the capability of LMs to classify on complex or specialized tasks is less well understood. We consider a highly complex task that is challenging even for humans: the classification of legal reasoning according to jurisprudential philosophy. Using a novel dataset of historical United States Supreme Court opinions annotated by a team of domain experts, we systematically test the performance of a variety of LMs. We find that generative models perform poorly when given instructions (i.e. prompts) equal to the instructions presented to human annotators through our codebook. Our strongest results derive from fine-tuning models on the annotated dataset; the best performing model is an in-domain model, LEGAL-BERT. We apply predictions from this fine-tuned model to study historical trends in jurisprudence, an exercise that both aligns with prominent qualitative historical accounts and points to areas of possible refinement in those accounts. Our findings generally sound a note of caution in the use of generative LMs on complex tasks without fine-tuning and point to the continued relevance of human annotation-intensive classification methods.

## 1 Introduction

Academia and industry increasingly use generative language models (LMs) for document annotation and class-prediction tasks, which promise enormous improvements in cost and efficiency. However, research tends to focus on relatively simple and generic annotation contexts, such as topic or query-keyword relevance (Gilardi et al., 2023; He et al., 2023; Törnberg, 2023). But many potential applications call for annotation or prediction of complex or specialized concepts, such as whether a writer reflects a particular school of thought. These questions may be difficult even to *describe* to a trained human annotator, much less apply. It is unclear if generative LMs perform well on this type of complex and specialized task.

In this study we systematically examine the ability of large LMs to parse a construct that is difficult even for highly trained annotators: modes of *legal reasoning*. We consider two prominent modes of legal reasoning that judges employ as identified by legal historians, in addition to a null or non-interpretative class. Although the *classes* of legal reasoning identified by historians reflect relatively well-defined concepts, determining whether a particular document reflects a mode of reasoning can be exceptionally challenging. We suspect this is common to many high-value but specialized tasks, such as classifying complex emotional states or detecting indirect racial or gender bias. These tasks often require both abstract reasoning and specialized knowledge.

Legal reasoning is a suitable setting for examining model performance on a highly complex classification task. The foundation of our research is a new dataset of thousands of paragraphs of historical Supreme Court opinions annotated by a team of upper-year students at a highly selective law school. We find that even the largest models perform poorly at the task without fine-tuning, even when using similar instructions as those given to human annotators. This finding suggests that LMs, even as augmented through few-shot or chain-of-thought prompting, may not be well-suited to complex or specialized classification tasks without task-specific fine-tuning. For such tasks, substantial annotation by domain experts remains a critical component.

To demonstrate this point, we examine the performance of established to cutting-edge LMs when fine-tuned on our annotated data. Our results show strong performance for many of these fine-tuned

models. Our analysis explores various approaches to model structure, such as a multi-class task versus serialized binary tasks, but we find that using an in-domain pre-trained model, LEGAL-BERT (Chalkidis et al., 2020), results in the highest performance for a task that requires specialized domain knowledge.

The primary contributions of this paper are as follows:

1. We develop a new dataset of domain-expert annotations in a complex area.
2. We find that SOTA in-context generative models perform poorly on this task.
3. We show that various fine-tuned models have relatively strong performance.
4. We study the relationship between our best-performing model's predictions and the consensus historical periodization of judicial reasoning, finding both substantial convergence and opportunities for refinement in the historical accounts.

In sum, our paper shows that in a complex and specialized domain, without fine-tuning, current generative models exhibit serious limitations; there is a continued need for domain-expert annotation, which can be effectively leveraged to unseen instances through fine-tuned models.[1]

## 2  Related Work

Researchers have developed strategies to guide LMs to perform complex tasks without the time and infrastructure costs of fine-tuning, often by breaking decisions down into multiple steps of reasoning. Wei et al. (2022) use few-shot chain-of-thought (CoT) prompting to provide a model with examples of intermediary logic before making a decision. An alternative, zero-shot CoT also results in improved performance in certain tasks, as LMs are prompted to break down their reasoning (e.g. "let's think step by step") (Kojima et al., 2023). Another procedure, Plan-and-Solve (PS) prompting, asks a model to devise and execute a plan for reasoning through a problem (Wang et al., 2023).

At certain tasks and with these prompting strategies, LMs perform annotation or classification tasks at the level of humans. Given the high costs (e.g., time, money, logistics) of collecting high-quality human-annotated data, recent work has suggested that annotation tasks previously performed by students, domain experts, or crowd-sourced workers could be replicated with equal performance by LMs. Generative models perform well on query-keyword relevance tasks (He et al., 2023); on topic detection in tweets (Gilardi et al., 2023); or on detecting political affiliation in tweets (Törnberg, 2023). Burnham (2023) suggests that zero-shot and few-shot models are a legitimate alternative for stance detection because of the unreliability of human annotators due to the vast contextual information annotators may or may not draw from. In the legal domain, scholars examine classification performance of generative LMs on the type of case (e.g., contracts, immigration, etc) (Livermore and Rockmore, 2023), or on the court's use of a specific canon in statutory construction (Choi, 2023).

The range of applications for which generative LMs might adequately perform is an open question. We have found limited work that requires specialized knowledge in addition to the use of abstract reasoning skills. In this study, we ask the models to engage in precisely this form of reasoning, which is challenging even for domain-expert humans. What distinguishes this form of reasoning is that it requires the analyst to conceptualize abstract principles and determine whether a specialized, domain-specific example fits one of those concepts. This difficulty contrasts with simpler tasks, which may key off well-established associations in training data between concepts, such as political affiliation and word usage.

## 3  Legal Reasoning

Our focus is on legal reasoning involving statutory interpretation.[2] In the United States, Congress writes statutes, but determining how statutes apply in individual cases is often left to the courts. Every year, the Supreme Court decides numerous cases of statutory interpretation, ranging from questions such as whether a tomato is a "vegetable" or a "fruit" for the purposes of import tariffs as in *Nix v. Hedden*,[3] to whether the Clean Air Act authorizes the Environmental Protection Agency to regulate greenhouse gases as in *West Virginia v. EPA*.[4]

---

[1]Code is available at: https://github.com/rosthalken/legal-interpretation

[2]Elsewhere, some of us examine jurisprudence more broadly (Stiglitz and Thalken, 2023).

[3]149 U.S. 304 (1893).

[4]142 S. Ct. 2587 (2022).

Jurists adopt a wide range of approaches to interpreting statutes and engaging in legal reasoning more generally. A classic distinction is between what 20th century legal scholar Karl Llewellyn referred to as "formal" and "grand" styles of reasoning (Llewellyn, 1960). Grand reasoning refers to a form of legal reasoning that respects precedent but is characterized by "the on-going production and improvement of rules which make sense on their face" (Llewellyn, 1960, p. 38). On interpretive questions, it therefore privileges work-ability, future orientation, and common-sense understandability. By contrast, formalism focuses not on the "policy" considerations of a law's consequences, but instead on its more mechanical application: "the rules of law are to decide the cases; policy is for the legislature, not for the courts... Opinions run in deductive form with an air or expression of single-line inevitability" (Llewellyn, 1960, p. 39).

Llewellyn's modes of legal reasoning apply more broadly than statutory interpretation. With respect to statutory interpretation specifically, under the grand style of reasoning "case-law statutes were construed 'freely' to implement their purpose, the court commonly accepting the legislature's choice of policy and setting to work to implement it" (Llewellyn, 1950, p. 400); showing his sympathies, under the formal style, Llewellyn wrote, "statutes tended to be limited or even eviscerated by wooden and literal reading, in a sort of long-drawn battle between a balky, stiff-necked, wrong-headed court and a legislature which had only words with which to drive that court" (Llewellyn, 1950, p. 400).

Though their terminology does not always follow Llewellyn, other legal scholars identify a similar primary distinction in legal reasoning. Horwitz, for instance, centers discussion on legal "orthodoxy," which seeks to separate law from consequences and elevate "logical inexorability" (Horwitz, 1992). Against orthodoxy, Horwitz identified a progressive critique, which "represented a broad attack on claims of Classical Legal Thought to be natural, neutral, and apolitical" (Horwitz, 1992, 189). Other prominent accounts follow Llewellyn's distinctions more explicitly (Gilmore, 2014). Operative doctrines in important areas of law, moreover, reflect the broad schools of thought: e.g., the "rule of reason" in anti-trust, which involves holistically examining the pros and cons of conduct rather than a rule-like test under the Sherman Act, may be understood to reflect the socially-aware grand school

| Class | Definition |
|---|---|
| Formal | A legal decision made according to a rule, often viewing the law as a closed and mechanical system. It screens the decision-maker off from the political, social, and economic choices involved in the decision. |
| Grand | A legal decision that views the law as an open-ended and on-going enterprise for the production and improvement of decisions that make sense on their face and in light of political, social, and economic factors. |
| None | A passage or mode of reasoning that does not reflect either the Grand or Formal approaches. Note that this coding would include areas of substantive law outside of statutory interpretation, including procedural matters. |

Table 1: Codebook definition for each class of legal reasoning.

of thought (Horwitz, 1992, 18).

Our contribution focuses on this broad consensus around a key distinction in the modes of legal reasoning. On the one hand, a mode of reasoning that is innovative, open-ended, and oriented to social, political, and economic consequences of law; on the other hand, a mechanical, logic-oriented approach that conceives of the law as a closed and deductive system of reasoning. Though scholars differ on terminology, we follow Llewellyn and refer to these schools as *Grand* and *Formal* (Table 1).

Not only does this basic conceptual consensus exist, but there is also rough consensus on periodization: that is, the periods of history in which each school was dominant. The "conventional" (Kennedy, 2006) view is that in the pre-Civil War period, the grand style dominated; in the period between the Civil War and World War I, the formal style dominated; the Grand school then dominated for much of the twentieth century (Llewellyn, 1960). The standard view is that we currently live in a period of formalism (Eskridge, 1990; Grey, 1999).

We use this periodization to validate our measure; but also use the measure to provide a nuanced account of historical trends in legal reasoning.

## 4 Data

We use a dataset of 15,860 historical United States Supreme Court opinions likely involving statutory interpretation and issued between 1870 and 2014.[5] The raw data come from Harvard's Caselaw Access Project.[6] Opinion text underwent minimal pre-processing, but all case citations were removed to reduce cognitive workload for the annotators.[7]

To create the dataset for annotation, we included only opinions that conduct statutory interpretation and then upsampled paragraphs likely to use formal or grand reasoning. The seed terms and details about this sampling procedure are included in Appendix A. In the final collection, 25% of paragraphs include at least one formal seed, 25% include at least one grand seed, and the remaining 50% are randomly sampled.

## 5 Human Annotations for Legal Reasoning

A team of domain experts, four upper-year law students at a highly selective law school, annotated selections from court opinions as formal, grand, or lacking statutory interpretation. This team collaboratively developed and tested a codebook (included in Appendix D) by iteratively annotating court opinions and calculating inter-rater reliability on a weekly basis over the spring 2023 semester.

The annotation task asked each annotator to assign one of three labels, "formal," "grand," or "none," to each paragraph. A fourth label, "low confidence," could be added in addition to one of the three core labels if the type of reasoning was ambiguous. We calculated inter-rater reliability using Krippendorff's *alpha* to evaluate agreement between the four labelers and across the three main classes. This coefficient was calculated weekly and guided the decision of when to start collecting data for training. Paragraphs with high disagreement were discussed in depth and these discussions led to the revision of our codebook.

We note that while this annotation is formally a three-way classification task, the low dimensional-

---

[5]This set of cases may include decisions on the merits and orders. See Appendix A for details on our opinion selection procedure.

[6]The raw data can be accessed here: https://case.law/. The Caselaw Access Project is not open-access but it grants unrestricted access to researchers.

[7]To do this, we used the eyecite Python library to identify the occurrence of case citations and replace them with the token '/[CITE/]' (Project, 2023).

ity of the output space does not imply that the task is easy. In fact, it took weeks for highly trained upper-year law students to reach a level of expertise at which they were able to reach consistent results.

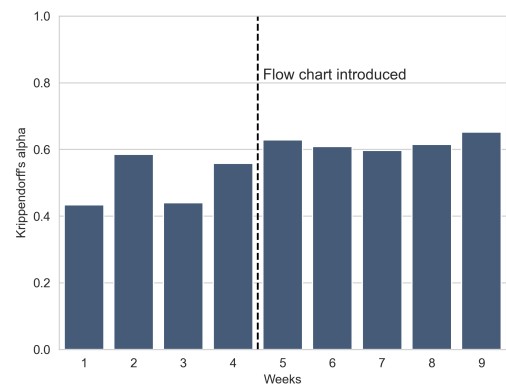

Figure 1: Weekly progression of Krippendorff's *alpha* for paragraphs and sentences assigned the formal, grand, and none labels. Decision chart added before Week 5.

Inter-rater reliability increased after the introduction of a decision chart (Figure 1), which broke down decisions about each of the classes into a series of guided questions (Appendix B). For each paragraph assigned the low confidence label, the team deliberated over possible labels until reaching a group decision through a majority vote. Training and evaluation data includes this resulting label for low confidence paragraphs; not the initial label.

In total, excluding paragraphs prior to decent inter-rater reliability, 2748 paragraphs were labeled and included in the training and evaluation data. Even with the upsampling of legal interpretation based on seed terms, paragraphs that did not engage in legal interpretation or interpreted something other than a statute, our "none" class, made up 68% of the data (Table 2). Grand reasoning was the second most common label, and formal the least common. Only 101 of these paragraphs received the additional low confidence label, and the formal class was the most common class to receive the low confidence label.

## 6 Automated Annotation with LMs

Though each member of the annotation team was an upper-year law student who had completed highly relevant coursework, the task remained difficult for the human annotators, as reflected in the mid-range inter-rater reliability (0.63 Krippendorff's *alpha*). The abstract concepts of the modes

| Class | # Ident'd | # LC | LC % |
|-------|-----------|------|------|
| Formal | 329 | 37 | 11.2% |
| Grand | 551 | 33 | 6.0% |
| None | 1869 | 31 | 1.7% |
| Total | 2748 | 101 | |

Table 2: Number of paragraphs fitting each class identified by annotators, and number assigned the low confidence (LC) class by its initial annotator.

of legal reasoning were clear, but determining whether specific instances reflected one mode or another required specialized knowledge and an ability to map those abstract concepts to the incomplete evidence in the paragraphs.

The complexity of this task makes it challenging for a generative model prompted in-context or with CoT reasoning. As an initial experiment, we begin with a slightly simplified task: identifying whether a passage involves *some* form of legal reasoning (regardless of class). We then compare a larger variety of models on the primary task of interest: identifying instances of formal and grand legal reasoning.

For both tasks, we compare the performance of in-context and fine-tuned models, with the expectation that identifying legal reasoning is more achievable for in-context models than identifying the specific formal or grand classes. Here, we test thresholds of task complexity, to better identify the point at which an annotated dataset for fine-tuning is needed; not just a carefully crafted prompt.

## 6.1 Model Training and Evaluation

In both tasks, we compare the performance of a set of fine-tuned models to a set of prompted models. Models were chosen based on established usage, popularity, and accessibility (i.e. model size), since applied NLP researchers may be less likely to have access to the computing power needed for extremely large models. The fine-tuned models include BERT-base (Devlin et al., 2019), DistilBERT (Sanh et al., 2020), and T5-small and T5-base (Raffel et al., 2020). We include one in-domain model, LEGAL-BERT-base, that was pre-trained from scratch on United States and European Union legal corpora, including United States Supreme Court cases (Chalkidis et al., 2020). Models prompted to identify legal reasoning include GPT-4 (OpenAI, 2023), FLAN-T5-large (Chung et al., 2022),

and Llama-2-Chat (7B) (Touvron et al., 2023). We created five random splits of the annotated data with 75% of the data in the training set and 25% of the data in the test segment. Models that were fine-tuned were fine-tuned over three epochs, with 50 warm-up steps, a learning rate of 2e-5, with a weight decay of 0.01.

## 6.2 Identifying Legal Reasoning

As a slightly simplified initial task, we begin by considering whether a model can detect instances in which some form of legal reasoning occurs (regardless of formal or grand reasoning). This remains a challenging task but is comparatively less complex than identifying the mode of reasoning. We consider any paragraph annotated as either formal or grand as being a paragraph where legal reasoning is present; this is a binary classification problem. We compare two procedures for identifying legal reasoning in text:

- In-context generative identification based on a description of legal reasoning (prompt included in Appendix C).
- Fine-tuned binary classification based on hand-labeled annotations.

All fine-tuned models perform relatively well on distinguishing paragraphs with legal reasoning from paragraphs without legal reasoning (Table 3). In comparison to these models, the zero-shot models prompted with a description of legal reasoning perform worse, and either over- or under-identify legal reasoning (e.g. high recall for the reasoning class but low precision). However, these models perform surprisingly well given the comparative workload behind each method: our fine-tuned models are built upon weeks of extensive labeling and discussion; the in-context models, only a prompt.

## 6.3 Identifying Types of Legal Reasoning

The primary task requires additional specialized knowledge in the identification of specific classes of reasoning, formal and grand. This task also requires the identification of imbalanced classes, as formal reasoning was only identified in 11% of all annotated paragraphs. We test various assemblies of models and compare fine-tuning with prompting for identifying legal reasoning in text. Our approaches to prompting include the following:

- *In Context, Descriptions*: An in-context prompt that provides the model with descriptions of the legal reasoning classes before

| Model | Macro | | | Interpretation | | | None | | |
|---|---|---|---|---|---|---|---|---|---|
| | F1 | P | R | F1 | P | R | F1 | P | R |
| *Fine-Tuned* | | | | | | | | | |
| BERT | 0.80 | 0.80 | 0.80 | 0.73 | 0.72 | 0.75 | 0.87 | 0.88 | 0.86 |
| LEGAL-BERT | **0.82** | **0.81** | **0.83** | **0.76** | **0.73** | 0.81 | **0.88** | **0.90** | 0.86 |
| DistilBERT | 0.80 | 0.80 | 0.80 | 0.73 | 0.72 | 0.74 | 0.87 | 0.88 | 0.86 |
| T5-base | 0.71 | 0.70 | 0.72 | 0.76 | 0.72 | 0.80 | 0.87 | 0.90 | 0.85 |
| T5-small | 0.75 | 0.75 | 0.75 | 0.67 | 0.66 | 0.67 | 0.84 | 0.84 | 0.83 |
| *In-Context* | | | | | | | | | |
| GPT-4 | 0.46 | 0.52 | 0.52 | 0.58 | 0.43 | **0.89** | 0.57 | 0.89 | 0.42 |
| FLAN-T5 | 0.42 | 0.54 | 0.50 | 0.04 | 0.41 | 0.02 | 0.80 | 0.68 | **0.98** |
| Llama-2-Chat | 0.16 | 0.34 | 0.56 | 0.44 | 0.33 | 0.67 | 0.03 | 0.68 | 0.02 |

Table 3: Model performance for binary interpretation averaged over 5 train test splits. Macro averages represent averages unweighted by class.

asking for inference on new paragraphs (Figure 2). The descriptions used in this prompt are the same presented to the annotation team in the codebook.

- *In Context, Examples*: An in-context prompt that provides the model with examples of the legal reasoning classes before asking for inference on new paragraphs (Appendix C). The examples used in this prompt are the same presented to the annotation team in the codebook.

- *Chain-of-Thought*: A CoT prompt that provides steps of reasoning to follow prior to determining the class of legal reasoning (Appendix C). The steps used in this prompt derive from the decision chart provided to annotators.

Each prompting strategy is derived from our codebook (see Appendix D), which guided human annotators through data annotation. We do not exhaustively explore prompts beyond our codebook. Instead, we consider whether a reasonable prompt that is successful for humans works well for a model. While it is possible that another, as-yet-unknown, prompt could have provided better results, we know that the language in our codebook is sufficient to describe the task and the desired results.

We contrast the results of the prompted generative models with the results from fine-tuned models. These models were fine-tuned with a variety of approaches, including:

- *Multi-Class*: A fine-tuned multi-class classi-

cation based on hand-labeled annotations.

- *Nested*: An assembly of models that breaks the classification task into nested binary stages. One model is fine-tuned to identify interpretation and another model to distinguish between grand and formal classes. The results from the first model are used by the second.

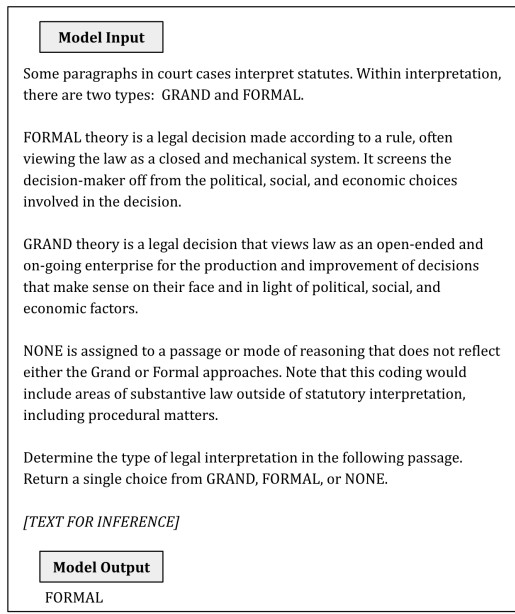

Figure 2: In-context prompt for identifying whether and which type of interpretation occurs.

## 7 Model Performance

We test the performance of all models on the same five test splits of data and find that the fine-tuned models consistently outperform the in-context mod-

| | Macro | | | Grand | | | Formal | | | None | | |
|---|---|---|---|---|---|---|---|---|---|---|---|---|
| Model | F1 | P | R | F1 | P | R | F1 | P | R | F1 | P | R |
| *Nested* | | | | | | | | | | | | |
| BERT-base | 0.67 | 0.67 | 0.68 | 0.65 | 0.62 | 0.68 | 0.49 | 0.50 | 0.48 | 0.87 | 0.88 | 0.86 |
| LEGAL-BERT | 0.69 | 0.68 | 0.71 | 0.67 | 0.64 | 0.71 | 0.54 | 0.51 | 0.57 | 0.88 | 0.90 | 0.85 |
| DistilBERT | 0.67 | 0.67 | 0.68 | 0.65 | 0.63 | 0.68 | 0.50 | 0.50 | 0.50 | 0.87 | 0.88 | 0.86 |
| T5-base | 0.67 | 0.67 | 0.68 | 0.66 | 0.59 | 0.75 | 0.47 | 0.52 | 0.43 | 0.87 | 0.90 | 0.85 |
| T5-small | 0.54 | 0.56 | 0.55 | 0.57 | 0.50 | 0.66 | 0.21 | 0.34 | 0.15 | 0.84 | 0.84 | 0.83 |
| *Multi-Class* | | | | | | | | | | | | |
| BERT-base | 0.68 | 0.68 | 0.68 | 0.67 | 0.65 | 0.69 | 0.50 | 0.52 | 0.48 | 0.88 | 0.88 | 0.87 |
| LEGAL-BERT | **0.70** | **0.70** | **0.71** | **0.68** | **0.65** | 0.72 | **0.55** | 0.55 | 0.55 | **0.88** | 0.89 | **0.87** |
| DistilBERT | 0.67 | 0.67 | 0.66 | 0.66 | 0.64 | 0.68 | 0.47 | 0.51 | 0.44 | 0.87 | 0.86 | 0.87 |
| T5-base | 0.64 | 0.64 | 0.65 | 0.66 | 0.63 | 0.69 | 0.51 | 0.52 | 0.50 | 0.87 | 0.88 | 0.86 |
| T5-small | 0.48 | 0.62 | 0.49 | 0.57 | 0.66 | 0.51 | 0.02 | 0.43 | 0.01 | 0.84 | 0.76 | 0.94 |
| *In-Context, Descriptions* | | | | | | | | | | | | |
| GPT-4 | 0.22 | 0.24 | 0.27 | 0.44 | 0.38 | 0.51 | 0.36 | 0.23 | 0.81 | 0.58 | 0.92 | 0.42 |
| FLAN-T5 | 0.19 | 0.36 | 0.36 | 0.36 | 0.24 | 0.77 | 0.16 | 0.11 | 0.28 | 0.04 | 0.72 | 0.02 |
| Llama-2-Chat | 0.20 | 0.34 | 0.36 | 0.35 | 0.22 | 0.92 | 0.00 | 0.00 | 0.00 | 0.24 | 0.82 | 0.14 |
| *In-Context, Examples* | | | | | | | | | | | | |
| GPT-4 | 0.45 | 0.47 | 0.54 | 0.45 | 0.38 | 0.57 | 0.36 | 0.25 | 0.65 | 0.62 | 0.86 | 0.48 |
| FLAN-T5 | 0.08 | 0.28 | 0.23 | 0.15 | 0.28 | 0.10 | 0.20 | 0.12 | 0.90 | 0.01 | 0.82 | 0.01 |
| Llama-2-Chat | 0.10 | 0.31 | 0.50 | 0.35 | 0.21 | **0.96** | 0.04 | 0.23 | 0.02 | 0.01 | 0.8 | 0.00 |
| *Chain-of-Thought* | | | | | | | | | | | | |
| GPT-4 | 0.34 | 0.37 | 0.37 | 0.25 | 0.50 | 0.17 | 0.43 | 0.32 | 0.67 | 0.78 | 0.80 | 0.76 |
| FLAN-T5 | 0.08 | 0.33 | 0.34 | 0.00 | 0.00 | 0.00 | 0.21 | 0.12 | **1.00** | 0.03 | 0.86 | 0.02 |
| Llama-2-Chat | 0.08 | 0.55 | 0.44 | 0.32 | 0.20 | 0.74 | 0.00 | **1.00** | 0.00 | 0.00 | **1.00** | 0.00 |

Table 4: Model performance averaged over five train test splits. Macro averages represent averages unweighted by class.

els (Table 4). Our results suggest that even state-of-the-art LMs may not be a suitable replacement for human annotation on highly complex and specialized classification tasks.[8]

Of all training or prompting procedures, models fine-tuned to perform multi-class classification tend to result in the highest performance. Out of all models, the best performing model is LEGAL-BERT, the one in-domain model included in this analysis. GPT-4 performs worse than all fine-tuned models, but has much better performance than Llama-2-Chat or FLAN-T5. Llama-2-Chat and FLAN-T5 greatly over-predict one of the three classes, and rarely predict the other two classes, making the

recall for one class artificially high.[9]

Also notable, the performance of the generative models on this more complex task is low compared to the simpler task of identifying whether *some* type of relevant legal reasoning occurs (Table 3). This is true both in absolute terms and relative to the in-domain, fine-tuned models. For instance, the macro F1 for GPT-4 on the simpler task is 0.46, 0.36 lower than the corresponding F1 for the in-domain, fine-tuned model. On this more complex task, the macro F1 for GPT-4 with descriptions is 0.22, 0.48 lower than the F1 for the in-domain, fine-tuned model.

---

[8]Our reported results employ a user-role and the default temperature on GPT-4. We experimented with zero-ed out temperature setting and with adding a system prompt, but the results did not improve substantially. Additionally, for the Llama-2-Chat models we used the same prompts as the other models but added the Llama-2-Chat-specific formatting that is necessary for instructing this model.

[9]We inspect the generated text from these models and find that FLAN-T5 and GPT-4 often over-predict certain classes, but these models rarely hallucinate or return text beyond the requested class (e.g. "grand"). Unsurprisingly, Llama-2-Chat often returns additional text beyond the class label; we extract the class label (if it occurs) from the text and use that as the label for evaluation.

# 8  Application: Periods of Legal Reasoning

The conventional wisdom among legal observers is that we currently live in a period in which the formal style of reasoning predominates (Eskridge, 1990; Grey, 1999). Yet it has not always been this way: in other historical periods, the grand style of reasoning prevailed. Indeed, there is a rough consensus in the legal literature regarding historical periodization (Kennedy, 2006).

Writing in the mid-twentieth century, Llewellyn identified three periods of legal reasoning. Prior to the Civil War, the grand style of reasoning predominated; from the Civil War to World War I, the formal style of reasoning prevailed; and from World War I onward, courts again operated under the grand style of reasoning. More recently, scholars identify the 1980s as a critical point of transition towards formalism (Eskridge, 1990). Other scholars identify fundamentally similar periodizations (Horwitz, 1992; Gilmore, 2014), and though differences exist, it is possible to speak of a "conventional" view (Kennedy, 2006). These historical characterizations arise from leading scholars reading judicial opinions and forming judgments through the use of their full faculties about the prevailing style of reasoning.

Our data starts at Reconstruction (the period following the US Civil War) and allows us to examine the convergence between the scholarly consensus historical periodization and the historical periodization implied by our LM-derived results. We can also use our predictions to offer more granular assessments of the periods and potentially to adjudicate differences among the views of prominent scholars. This latter analysis is preliminary, in part, because earlier scholars examined judicial reasoning broadly, whereas our current analysis considers only Supreme Court opinions involving statutory interpretation.[10]

For this exercise, we study historical trends in the predictions from the highest performing model, multi-class, fine-tuned LEGAL-BERT. We examine yearly averages at both the paragraph level and the opinion level.[11] We focus only on paragraphs

that involve interpretation, and code paragraphs classified as "formal" with a 1 and paragraphs classified as "grand" with a 0. These yearly averages, therefore, reveal the proportion of interpretive paragraphs that classify as formal as opposed to grand. Figure 3 plots the yearly averages over our series: the left panel (panel a.) shows the yearly average at the paragraph level, and the right panel (panel b.) aggregates paragraphs within documents to show the yearly average at the opinion level.

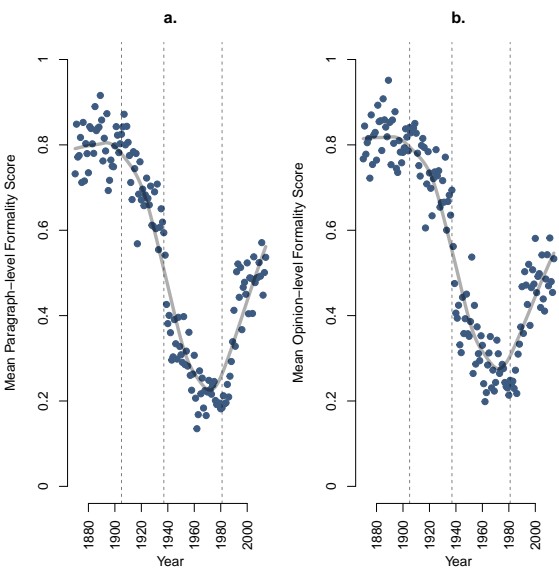

Figure 3: Historical trends in interpretive style at the paragraph (panel a.) and document (panel b.) levels, by year. Coded so that 1=formal and 0=grand.

Broadly understood, the historical trends in our predictions converge with the views of Llewellyn and other legal observers. That is, the period after the Civil War and before World War I was characterized by formal judicial reasoning; the mid-century was characterized by grand legal reasoning; and we now live in a period of formalist resurgence. The story is essentially the same at the paragraph or the document level.

But our predictions also allow for a more granular assessment of the historical periods. To illustrate this, we use dashed vertical lines in Figure 3 to denote important historical events: in 1905, the Supreme Court decided *Lochner v. New York*,[12] which some observers note as a highwater point for formalism; in 1937, a year of "judicial revolution," in which the Supreme Court is widely viewed to have shifted its jurisprudence from opposition to ac-

---

[10]We screen opinions for these predictions using the statutory interpretation filter identified in Appendix A. Stiglitz and Thalken (2023) provide a more comprehensive LM analysis of historical trends in jurisprudence.

[11]An opinion-level prediction represents the average of paragraphs in that opinion. If the number of paragraphs in opinions is not time invariant, historical trends in opinions may not be

the same as trends in paragraphs.

[12]198 U.S. 45 (1905).

ceptance of federal and state regulations;[13] and in 1981, as President Reagan's judicial appointments started to take office and a possible marker for the formalist revival.[14]

To a striking degree, these historical markers correspond with changes in our metric of jurisprudence. Consistent with those who see *Lochner* as a high-water point for formalism, the prevalence of formalist reasoning declines after 1905. Likewise, we see a remarkable increase in the prevalence of grand reasoning in 1937. This pattern is consistent with a "judicial revolution" in jurisprudence to accommodate regulatory programs, the type of which had been earlier invalidated under formalist regimes. Finally, our measures recover a sharp increase in formalism in the 1980s, again consistent with the views of legal observers.

These results represent some of the first long-run quantitative characterization of trends in jurisprudential philosophies. They both broadly support the qualitative characterizations of legal scholars and provide opportunities for refinement of legal theory and historical accounts.

## 9 Conclusion

We found that for a task involving abstract reasoning in addition to specialized domain-specific knowledge, it remains essential to have an annotated dataset created by domain experts. Although other work has shown that generative models are able to replicate annotation for complex tasks using carefully crafted prompts, we demonstrate that models fine-tuned on a sizable dataset of expert annotations perform better than models instructed to perform the task through in-context and CoT prompts. We recommend that researchers use caution when employing non-fine-tuned generative models to replicate complex tasks otherwise completed by humans or with human supervision. Best practices would call for human validation of generative model results and an assessment of cost-performance tradeoffs with respect to in-domain models.

---

[13]This revolution is also known as the "switch in time that saved nine," referring to the changed voting behavior of Justice Owen Roberts in response to the running threat by President Roosevelt to pack the Court.

[14]Justice Scalia, for instance, was appointed by President Reagan to the Supreme Court in 1986, and is often viewed as the single most influential person in the rise of new formalism. For an account of that rise, see Eskridge (1990).

## 10 Limitations

A limitation of this study is the relatively low inter-rater reliability between annotators even after extensive training and conversation. This relatively low reliability results from the difficulty of the task and the inevitable ambiguity of some passages, especially when read out of case context. Another limitation relates to our prompting strategy: to make the in-context prompting more comparable to working with the team of annotators, we use the codebook descriptions and examples in the in-context prompts. Likely, these descriptions and examples could have been optimized for better model performance through additional prompt strategies, and our results for these models may depict lower performance than is possible.

## Acknowledgments

We thank our annotation team, including Houston Brown, Michael Demers, Sarah Engell, and Josiah Rutledge, for their extensive work creating this dataset. We also thank Zach Clopton, Vijay Karunamurthy, Gregory Yauney, Andrea Wang, Federica Bologna, Anna Choi, Rebecca Hicke, Kiara Liu, and participants at workshops at Cornell Law School and Cornell Computer Science Department for helpful comments. This work was supported by NSF #FMiTF-2019313 and NSF #1652536.

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

# A    Selection of Paragraphs for Labeling

We use the following seed terms to up-sample paragraphs that more likely reflect the interpretive approaches of interest.[15]

---

[15]These terms largely draw from earlier efforts by legal scholars to develop search terms that recover cases involving

- **Grand seeds:** conference report, committee report, senate report, house report, assembly report, senate hearing, house hearing, assembly hearing, committee hearing, conference hearing, floor debate, legislative history, history of the legislation, conference committee, joint committee, senate committee, house committee, assembly committee, legislative purpose, congressional purpose, purpose of congress, purpose of the legislature, social, society

- **Formal seeds:** dictionary, dictionarium, liguae britannicae, world book, funk & wagnalls, expressio, expresio, inclusio, noscitur a sociis, noscitur a socis, ejusdem generis, last antecedent, plain language, whole act, whole-act, whole code, whole-code, in pari materia, meaningful variation, consistent usage, surplusage, superfluit, plain meaning, ordinary meaning, word

The selection of paragraphs to annotate occurred through a series of steps:

1. We include only opinions that perform statutory interpretation. We identify these opinions by finding opinions that include any of the tokens 'statute', 'legislation', or 'act', within 200 characters of the tokens 'mean', 'constru' (i.e. construct), 'interpret', 'reading', or 'understand'.

2. Opinions that pass the statutory interpretation filter were split into paragraphs. In each paragraph, we looked for the occurrence of different seed terms corresponding to either formal or grand reasoning.

3. Of the total number of paragraphs used for labeling, 25% included one or more formal seeds, 25% included one or more grand seeds, and 50% included none of the seed terms. This proportion remained the same until the last two rounds of labeling when more examples of formal or grand seeds were included. During those two rounds, the proportion of formal and grand seeds was increased to 40% for both classes.

---

methods of statutory interpretation related to the formal and grand styles of jurisprudence. Choi (2020) helpfully collects many of these terms; see also related efforts (Staudt et al., 2005; Calhoun, 2014; Bruhl, 2018).

## B  Decision Chart

A decision chart was created and provided to annotators between the fourth and fifth weeks of annotations (Figure 4). Following the incorporation of this decision chart, we saw boosted inter-rater reliability and more consistent agreement between annotators.

## C  Prompts

We designed three prompting strategies to instruct LMs to identify legal interpretation and classes of legal interpretation in text. These prompts are included in Figures 2 and 5. All prompts were modeled after our annotation codebook.

## D  Codebook

The codebook was iteratively created throughout the process of annotation to guide annotators. Table 5 includes the final definitions of each class alongside core examples of each class.

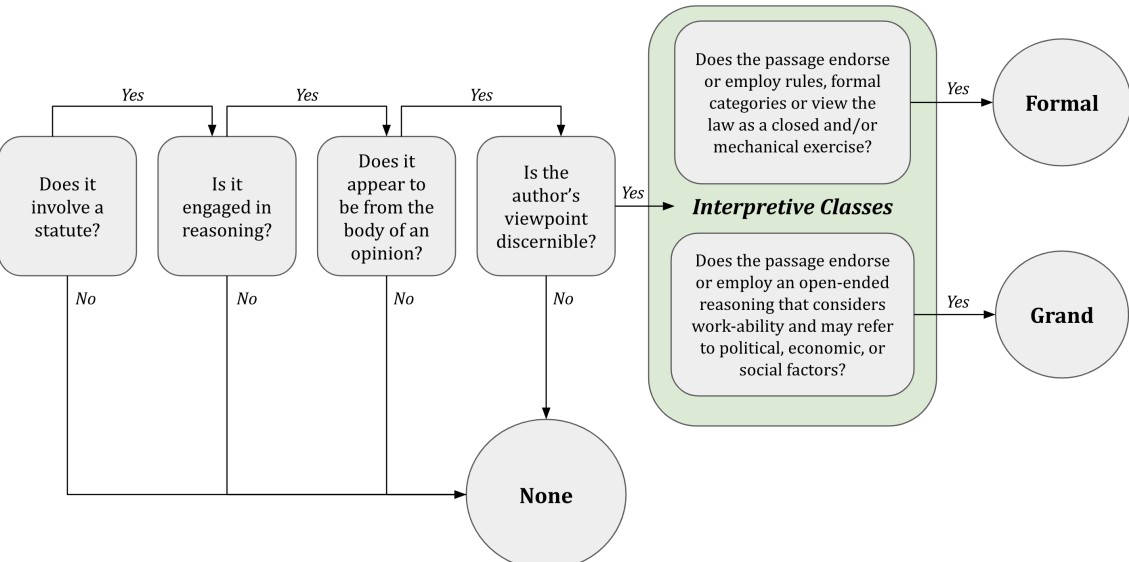

Figure 4: The decision chart provided to the annotation team.

**Model Input**

Some paragraphs in court cases interpret statutes. In this type of paragraph, there is an analysis of a statute and a claim made about its meaning.

In the following paragraph, determine if legal interpretation occurs. If yes, respond with "interpretation" and if not, respond with "no interpretation."

*Nevertheless, respondent urges that the legislative purpose of the statute is best served by construing it to permit some choice in determining the length of the penalty period. In respondent's view, the purpose of the statute is essentially remedial and compensatory, and thus it should not be interpreted literally to produce a monetary award that is so far in excess of any equitable remedy as to be punitive.*

**Model Output**

interpretation

a.

**Model Input**

Some paragraphs in court cases interpret statutes. Within interpretation, there are two types: grand and formal.

Grand interpretation represents a legal decision that views law as an open-ended and on-going enterprise for the production and improvement of decisions that make sense on their face and in light of political, social, and economic factors.

Formal interpretation is a legal decision made according to a rule, often viewing the law as a closed and mechanical system. It screens the decision-maker off from the political, social, and economic choices involved in the decision.

Let's analyze the following passage step-by-step. First, determine if it interprets a statute. Second, if it interprets a statute, determine whether the interpretation is grand or formal. The first word in your response should label the passage with "GRAND", "FORMAL", or "NONE" and then explain why you chose that label.

*[TEXT FOR INFERENCE]*

**Model Output**

FORMAL

b.

**Model Input**

Determine the legal interpretation used in the following passage. Return a single choice from GRAND, FORMAL, or NONE. Here are examples:

###
Text: [FORMAL CODEBOOK EXAMPLE]
FORMAL

###
Text: [GRAND CODEBOOK EXAMPLE]
GRAND
###

###
Text: [NONE CODEBOOK EXAMPLE]
NONE

###
Text: [TEXT FOR INFERENCE]
###

**Model Output**

GRAND

c.

Figure 5: In-context prompt for identifying whether interpretation occurs or not. Prompt a is the prompt used for binary interpretation. Prompt b is the prompt used for CoT reasoning and the classes of legal interpretation. Prompt c is the prompt used for few-shot classification of the classes of legal interpretation.

| Class | Definition | Example |
|---|---|---|
| Formal | A legal decision made according to a rule, often viewing the law as a closed and mechanical system. It screens the decision-maker off from the political, social, and economic choices involved in the decision. | Accepting this point, too, for argument's sake, the question becomes: What did "discriminate" mean in 1964? As it turns out, it meant then roughly what it means today: "To make a difference in treatment or favor (of one as compared with others)." Webster's New International Dictionary 745 (2d ed. 1954). To "discriminate against" a person, then, would seem to mean treating that individual worse than others who are similarly situated. [CITE]. In so-called "disparate treatment" cases like today's, this Court has also held that the difference in treatment based on sex must be intentional. See, e.g., [CITE]. So, taken together, an employer who intentionally treats a person worse because of sex—such as by firing the person for actions or attributes it would tolerate in an individual of another sex—discriminates against that person in violation of Title VII. Bostock v. Clayton County |
| Grand | A legal decision that views the law as an open-ended and on-going enterprise for the production and improvement of decisions that make sense on their face and in light of political, social, and economic factors. | Respondent's argument is not without force. But it overlooks the significance of the fact that the Kaiser-USWA plan is an affirmative action plan voluntarily adopted by private parties to eliminate traditional patterns of racial segregation. In this context respondent's reliance upon a literal construction of §§ 703 (a) and (d) and upon McDonald is misplaced. See [CITE]. It is a "familiar rule, that a thing may be within the letter of the statute and yet not within the statute, because not within its spirit, nor within the intention of its makers." [CITE]. The prohibition against racial discrimination in §§ 703 (a) and (d) of Title VII must therefore be read against the background of the legislative history of Title VII and the historical context from which the Act arose. See [CITE]. Examination of those sources makes clear that an interpretation of the sections that forbade all race-conscious affirmative action would "bring about an end completely at variance with the purpose of the statute" and must be rejected. [CITE]. See [CITE]. Steelworkers v. Weber |
| None | A passage or mode of reasoning that does not reflect either the Grand or Formal approaches. Note that this coding would include areas of substantive law outside of statutory interpretation, including procedural matters. | The questions are, What is the form of an assignment, and how must it be evidenced? There is no precise form. It may be. by delivery. Briggs v. Dorr, CITE, citing numerous cases; Onion v. Paul, 1 Har. & Johns. 114; Dunn v. Snell, CITE; Titcomb v. Thomas, 5 Greenl. 282. True, it is said it must be on a valuable consideration, with intent to transfer it. But these last are requisites in all assignments, or transfers of securities, negotiable or not. It may be by writing under seal, by writing without seal, by oral declarations, accompanied in all cases by delivery, and on a just consideration. The evidence may be by proof of handwriting and proof of. possession. It may be proved by proving the signature of the payee or obligee on the back, and possession by a third person. 3 Gill & Johns. 218. |

Table 5: Codebook definition and examples of each of the interpretive classes.