# OpenReview forum: "Modeling Legal Reasoning: LM Annotation at the Edge of Human Agreement"
_EMNLP/2023/Conference — EMNLP 2023 Main_

### Official Review · Reviewer_TywZ · 2023-07-28

**Typos Grammar Style And Presentation Improvements:** Style is up to standard and clear.
**Soundness:** 4

**Excitement:**

4: Strong: This paper deepens the understanding of some phenomenon or lowers the barriers to an existing research direction.

**Missing References:**

None that I can think of.

**Paper Topic And Main Contributions:**

The paper makes an important contribution to the topic of automatically retrieving types of legal argument regarding statutory interpretation (mainly in the context of US Supreme Court judgments). It tests a range oif techniques and compes up with innovative ways of zooming in on a particularly efficient one such range of techniques.

**Questions For The Authors:**

Question 1A: Why not distinguish between 'literal' versus 'purposive' interpretation instead of 'grand' versus 'formal'?
Question 1B: If you had distinguished in that way, could the prompt provided have provided better results in the generative NLP experiments?

**Reasons To Accept:**

There are several reasons to accept the paper. First, it makes a contribution in a previously underexplored area. Second, it assesses various different ways of resolving the task that it sets out and compares these well. Third, it contains a helpful (but see below) conceptual framework distinguishing between different approaches to statutory interpretation.

**Reasons To Reject:**

There are no real reasons to reject the paper, but merely a suggestion that its conceptual framework could have been somewhat tighter. First, the authors' idea of 'intepretation' is noit sufficiently rigorous and clear (from the point of view of a legal scholar like myself). Whether an item falls within the extension of a legal concept is usually called a question of 'application' not of 'interpretation'. Interpretation means simply spelling out the linguistic meaning of an item (a statutory provisionsay) bu using paraphrases in clauses such as 'X means Y'. Second, and more importantly, it is not clear at all that Llewellyn's distinction between 'grand' and 'formal' styles captures something specific. In fact, it seems to me that it would have been much better to simply distinguish, as we do in some civil law systems, between 'literal' and 'purposive' interpretation of legal texts (such as statutes), the first being mainly geared towards the conventional linguistic meaning of the expression and the second towards identifying the purposes of the provision, i.e. the goals that the rule is supposed to serve. But again, these are not reasons to reject the paper, merely suggestions about strengthening it.

**Reproducibility:**

5: Could easily reproduce the results.

**Reviewer Confidence:**

3: Pretty sure, but there's a chance I missed something. Although I have a good feel for this area in general, I did not carefully check the paper's details, e.g., the math, experimental design, or novelty.

---

> ### Author Rebuttal · Authors · 2023-08-29
>
> Thank you for the thoughtful review! We appreciate your comments and suggestions on how to refine the paper.
>
> *Concern: Why not distinguish between 'literal' versus 'purposive' interpretation instead of 'grand' versus 'formal'?*
>
> This is a great question – in our revisions, we will draw parallels to these modes of interpretation. Though the “literal” and “purposive” distinction is closely related to the “formal” and “grand” distinction, we believe that the formal and grand distinction encompasses a somewhat wider range of reasoning methods. For instance, one aspect of formal reasoning, according to Llewellyn and others, is that it is closed off to social, political, and economic factors that operate at the time of the legal decision. These factors often but not always relate to the purpose of the law at the time of passage. Our weekly discussions with the annotation team were not uncommonly related to the degree to which a legal passage embraced social considerations, and we likewise attempted to integrate that factor into our prompts.
>
> *Concern: If you had distinguished in that way, could the prompt provided have provided better results in the generative NLP experiments?*
>
> A challenge with this research project is that we, in effect, find a “null” result for the generative language models. That is, we find that they do not perform adequately. A generic concern with null results is that they reflect an aspect of the study design rather than the data generating process: for instance, the study may be under-powered, or perhaps it focuses on the wrong place or time for an effect. Applied to our study, a concern is that the generative language models might perform better with different prompts—that is, perhaps our null is a function of our prompts rather than the generative models.
>
> We attempt to address this concern in three ways. First, we explore a range of prompt methodologies, including zero shot, few shot, and chain-of-thought approaches. None of these approaches results in strong performance. Second, thematically, our interest is in the ability of the generative models to replace human inputs in complex research tasks. Our general approach, therefore, is to provide the models with similar types of instructional context that we provided to our annotation team. Finally, we acknowledge that we cannot completely dispel the idea that generative models, with different prompts, would perform better. Viewed broadly, we see our paper as advancing a debate rather than settling it. In revision, we adapt language to reflect these points and limits to the study.

---

### Official Review · Reviewer_WWog · 2023-08-02

**Soundness:** 4

**Excitement:**

4: Strong: This paper deepens the understanding of some phenomenon or lowers the barriers to an existing research direction.

**Paper Topic And Main Contributions:**

The authors proposed a legal classification dataset and benchmarked LMs task-solving and reasoning performance off the shelf. The results justified generic LMs have limited ability to solve specialized tasks without task-specific fine-tuning.

**Questions For The Authors:**

A. Can you justify the reason for choosing BERT, DistilBERT as baselines, not more recent pre-trained SOTA models such as T5?

B. Follow-up question, why choose FLAN-T5-Large instead of FLAN-T5-XL and XXL, is that due to computing resource limitations?

C. Speaking of FLAN-T5-Large, the results are abnormally low. My personal experience with FLAN-T5 is that the model may generate text that is different from the prompted options, and you need to write some heuristics to map them back (for a faithful performance evaluation). Have you checked what FLAN-T5 is outputting?

D. What is the human performance in this dataset?  If the inter-rater consensus is low, what did you do to verify the annotation? e.g., hiring more people for majority voting. In addition, if the human performance is lower than fine-tuned model performance, will it hurt the annotation quality and dataset validity?

**Reasons To Accept:**

The dataset seems of relatively good quality, which is domain-experts annotated. The experiment setup and results are valid.

**Reasons To Reject:**

The primary concern is the novelty (or the position/framing of this paper).

- More specifically, if this is a resource paper, The pro of this dataset is expert annotation and well-defined task, however, I would say the difficulty of this dataset is not hard enough, where a fine-tuned LEGAL-BERT can already reach over 80% accuracy, and I would assume a much larger model can perform even better when fine-tuned, so the space left for improvement is very limited. In addition, the authors spend a lot of space introducing and describing the task from the legal history perspective, I would suggest shortening that part in the main content and putting the story in the appendix.

- If this is a paper focusing on the benchmarking of SOTA generic LMs, the experiments are not comprehensive to provide enough contribution.

The authors only conduct fine-tune experiments on relatively small (and a bit early) LMs such as BERT and DistilBERT, but do not include seq2seq models (e.g., BART, T5) and even larger models (e.g., LLaMA), which lead to the potential issue that the baselines are not SOTA.

For the no-tuning experiment, the authors only compared GPT-4 and FLAN-T5-large(why not FLAN-T5-XL and XXL?, or LLaMA and Alpaca?), and I do leave a question below for FLAN-T5's result here. The prompts in the Appendix are okay, but there are many questions left (if the author wants to comprehensively support the claim):
How to justify the prompt you designed is making the best of the models?
How do you determine the number of few-shot examples, and how do you select them?
How to justify the few-shot setting you use is making the best of the models?

To summarize, I appreciate the careful design of the dataset collection and experiment setup & analysis. However, I would suggest the authors sell the paper in a way that emphasizes all the pros (and avoid the cons), instead of trying to explore everything but not doing everything comprehensively.
This paper is of a relatively higher quality compared with all the papers I reviewed for EMNLP this year (maybe it's just my sampling bias). As a result, I do have higher expectations for the authors to make this work outstanding and solid. I may change my score depending on the rebuttal.

ps. I recently found one paper that is exploring using LLM in another specialized domain, and I would recommend the authors take a look at how extensive their experiments are to support their similar claim (LLMs are not suitable for specialized domains off the shelf).
Let me make my point clear, I'm not suggesting the authors cite this paper (it was just submitted to Arxiv a couple of days ago and of a different domain), but I'd like to use it as an example to show the level of comprehensiveness you need, if you want to emphasize the contribution on the benchmarking experiment side. They explored a lot of different prompt designs for zero-shot, conducted few-shot experiments, conducted subsampling and transfer learning experiments for fine-tuning the LM, and used many SOTA models (Alpaca, Lora-finetuned Alpaca, GPT-3.5, in addition to BERT and Mental-RoBERTa) for comparison.
- Leveraging Large Language Models for Mental Health Prediction via Online Text Data (https://arxiv.org/abs/2307.14385)

**Reproducibility:**

4: Could mostly reproduce the results, but there may be some variation because of sample variance or minor variations in their interpretation of the protocol or method.

**Reviewer Confidence:**

4: Quite sure. I tried to check the important points carefully. It's unlikely, though conceivable, that I missed something that should affect my ratings.

**Typos Grammar Style And Presentation Improvements:**

line 355, 'all models were tested with the same subset of training data', should it be 'testing data'?

---

> ### Author Rebuttal · Authors · 2023-08-29
>
> Thank you for the thoughtful review! We appreciate your concern for keeping all EMNLP submissions at a high quality and your recommendations for ways to improve our experimentation.
>
> *Concern: positioning of the work; “is this a resource paper or is this a benchmark paper?”*
>
> We recognize that the resource paper and the model comparison paper are two common archetypes, but we view this paper as falling into another category of EMNLP submissions. In this case our research question is "what happens when we apply LLMs to a classification task that highly-trained humans find difficult?" As an NLP Applications paper, we believe that answering this question requires a hybrid approach. This approach includes the influence of domain-specific details on data preparation and the careful selection of models to rule out the possibility that observations hold for only one model.
>
> *Concern (A): the experimentation is not comprehensive enough; “Can you justify the reason for choosing BERT, DistilBERT as baselines, not more recent pre-trained SOTA models such as T5?”*
>
> We agree with this concern, and will ensure that the quality of the model performance testing matches the quality of our dataset creation, including more examples of how various models perform on this task. We understand the concern about the lack of justification for the models selected, and will clarify our choices for each model. Our criteria for inclusion are that models should be commonly used for applied NLP research, that they can be fine-tuned for "instruction" style interaction, and we prioritized models that are either available through public APIs or small enough to be run by domain experts without access to their own GPU hardware.
>
> To provide a more complete assessment of available LLMs, we have begun testing the performance of additional models (T5-base, T5-small, and Llama-2-7b-chat) noted in the table below. Model names that are bolded are new models; the rest of the results are the same as the submitted version. Thus far, the results from these models align with our prior experiments. We plan to test Llama-2-7b-chat in the chain-of-thought and few-shot setting and T5-base and T5-small with nested fine-tuning, but were unable to complete this before the rebuttal deadline due to time constraints.
>
> |                          | Macro Averages |          |          | Grand    |          |          | Formal   |          |          | None     |          |          |
> |--------------------------|----------------|----------|----------|----------|----------|----------|----------|----------|----------|----------|----------|----------|
> |                          | F1             | P        | R        | F1       | P        | R        | F1       | P        | R        | F1       | P        | R        |
> | Nested                   |                |          |          |          |          |          |          |          |          |          |          |          |
> | BERT-base                | 0.67           | 0.67     | 0.68     | 0.65     | 0.62     | 0.68     | 0.49     | 0.50     | 0.48     | 0.87     | 0.88     | 0.86     |
> | LEGAL-BERT-base          | 0.69           | 0.68     | 0.71     | 0.67     | 0.64     | 0.71     | 0.54     | 0.51     | 0.57     | 0.88     | 0.90 &   | 0.85     |
> | DistilBERT               | 0.67           | 0.67     | 0.68     | 0.65     | 0.63     | 0.68     | 0.50     | 0.50     | 0.50     | 0.87     | 0.88     | 0.86     |
> |                          |                |          |          |          |          |          |          |          |          |          |          |          |
> | Multi-Class              |                |          |          |          |          |          |          |          |          |          |          |          |
> | BERT-base                | 0.68           | 0.68     | 0.68     | 0.67     | 0.65     | 0.69     | 0.50     | 0.52     | 0.48     | 0.88     | 0.88     | 0.87     |
> | LEGAL-BERT-base          | **0.70**       | **0.70** | **0.71** | **0.68** | **0.65** | 0.72     | **0.55** | **0.55** | 0.55     | **0.88** | 0.89     | **0.87** |
> | DistilBERT               | 0.67           | 0.67     | 0.66     | 0.66     | 0.64     | 0.68     | 0.47     | 0.51     | 0.44     | 0.87     | 0.86     | 0.87     |
> | **T5-base**              | 0.64           | 0.64     | 0.65     | 0.66     | 0.63     | 0.69     | 0.51     | 0.52     | 0.50     | 0.87     | 0.88     | 0.86     |
> | **T5-small**             | 0.48           | 0.62     | 0.49     | 0.57     | 0.66     | 0.51     | 0.02     | 0.43     | 0.01     | 0.84     | 0.76     | 0.94     |
> |                          |                |          |          |          |          |          |          |          |          |          |          |          |
> | In-Context, Descriptions |                |          |          |          |          |          |          |          |          |          |          |          |
> | GPT-4                    | 0.22           | 0.24     | 0.27     | 0.44     | 0.38     | 0.51     | 0.36     | 0.23     | **0.81** | 0.58 &   | **0.92** | 0.42     |
> | FLAN-T5-large            | 0.19           | 0.36     | 0.36     | 0.36     | 0.24     | 0.77     | 0.16     | 0.11     | 0.28     | 0.04     | 0.72     | 0.02     |
> | **Llama-2-7b-chat**      | 0.2            | 0.34     | 0.36     | 0.35     | 0.22     | **0.92** | 0        | 0        | 0        | 0.24     | 0.82     | 0.14     |
> |                          |                |          |          |          |          |          |          |          |          |          |          |          |
> | In-Context, Examples     |                |          |          |          |          |          |          |          |          |          |          |          |
> | GPT-4                    | 0.45           | 0.47     | 0.54     | 0.45     | 0.38     | 0.57     | 0.36     | 0.25     | 0.65     | 0.62     | 0.86     | 0.48     |
> | FLAN-T5-large            | 0.08           | 0.30     | 0.23     | 0.14     | 0.27     | 0.10     | 0.20     | 0.11     | 0.90     | 0.01     | 0.92     | 0.01     |
> |                          |                |          |          |          |          |          |          |          |          |          |          |          |
> | Chain-of-Thought         |                |          |          |          |          |          |          |          |          |          |          |          |
> | GPT-4                    | 0.34           | 0.37     | 0.37     | 0.25     | 0.50     | 0.17     | 0.43     | 0.32     | 0.67     | 0.78     | 0.80     | 0.76     |
> | FLAN-T5-large            | 0.08           | 0.33     | 0.34     | 0.00     | 0.00     | 0.00     | 0.21     | 0.12     | 1.00     | 0.03     | 0.88     | 0.02     |
>
>
> *Concern (B): why choose FLAN-T5-Large instead of FLAN-T5-XL and XXL, is that due to computing resource limitations?*
>
> We limit to a set of slightly smaller models due to resource constraints and our substantive criteria for model inclusion. While larger models like FLAN-T5-XL or XXL may have better results, these models are less accessible to the average academic researcher.
>
> *Concern: How do you justify the prompt you designed is making the best of the models?*
>
> We agree that there may be  prompts that would improve the results of the generative models. As we observe more fully in our response to reviewer TywZ, we try to address this concern by exploring a range of prompt methodologies, by emphasizing that our interest is in the capacity of the generative models to replace human inputs in complex research tasks, and by being forthright about the limits of our study.
>
> Our general view is that the paper advances the debate but does not settle it. We will revisit the language in the paper to conform any errant language to reflect this viewpoint. For instance,  we can clarify that we are not evaluating a stronger hypothesis ("there does not exist a prompt that is effective") but rather a weaker hypothesis ("a reasonable prompt that is successful for humans does not work as well").
>
> *Concern (C): Have you checked what FLAN-T5 is outputting?*
>
> Yes, we have verified that these results are correct. For the descriptive prompt, where FLAN-T5 performs slightly better, FLAN-T5 rarely identifies a paragraph as “None” and overpredicts Grand and Formal. For the few-shot and chain-of-thought prompts, FLAN-T5 almost always predicts a paragraph as Formal.
>
> *Concern (D): What is the human performance in this dataset? If the inter-rater consensus is low, what did you do to verify the annotation? In addition, if the human performance is lower than fine-tuned model performance, will it hurt the annotation quality and dataset validity?*
>
> This is a complex task, requiring extensive legal knowledge and careful analysis of each passage. We see the final inter-rater reliability (0.63 Krippendorff’s *alpha*) as an appropriate score, relative to task difficulty.
>
> We verified the annotations in multiple ways. First, we leveraged the knowledge of the group of four law students who had completed the annotation. Each weekly batch of paragraphs for annotation contained a small overlapping set of paragraphs that were assigned to all four annotators. During our weekly discussions, each annotator justified the class assigned to difficult paragraphs in this shared sample. As a group, we then deliberated on that class, and made changes to the codebook based on that discussion.
>
> Even after extensive training, discussion, and deliberation, there were a set of paragraphs that remained ambiguous as to classification. However, we see these examples as highly important to include in the training data, and so we treated them with additional care. Each annotator could assign a “low confidence” score to a paragraph, which then resulted in a group deliberation and vote about what the paragraph’s correct assignment should be. The label that came from this majority vote was included in the training and validation sets.
>
> We further undertook efforts to validate the predictions from our preferred model, including several exercises not reported in the current draft but that may be in the revised draft. For instance, our annotation codebook includes paragraph exemplars for each class. We examine the case-level predictions for the parents of those paragraphs, and find that the model predicts case-level classes as expected (e.g., the median formal example parent case contained about 60 percent formal reasoning in its paragraphs; the corresponding figure for the grand examples was about 25 percent). In another exercise, we examine the formality of opinions written by different authors and find, as expected, that jurists known for formal jurisprudence, such as Justice Scalia, return with high average scores.
>
> Finally, the current draft of the paper includes one of our validation efforts, the results reported in section 8 (“Periods of Legal Reasoning”). That section shows the yearly averages for formality in legal reasoning over a roughly 150 year period. While we can expect some degree of misalignment between qualitative narratives of historical trends in legal reasoning, the fundamental convergence between the qualitative historical consensus and our own findings strongly suggests that our annotations correspond with the target concepts.

---

### Official Review · Reviewer_ikuS · 2023-08-04

**Soundness:** 5

**Excitement:**

4: Strong: This paper deepens the understanding of some phenomenon or lowers the barriers to an existing research direction.

**Paper Topic And Main Contributions:**

This paper develops a large annotated database of legal reasoning (specifically, the data is examples of "formal" and "grand" styles of legal reasoning in court decisions.  Using this data, the paper then compares classification performance across several models.  The conclusion is that for current models, the classification task is challenging.  Most interesting (at least to me) was the time-series analysis of court cases that quantifies the shift in opinion style.

**Questions For The Authors:**

(perhaps these are questions for a follow up paper.)

A:  The mean scores presented in figure 3 are interesting.  Is there any interesting time series in the cross-section?  For example, perhaps is a plot of variance (st.dev) across time.
B:  In the time-series plot in Figure 3, is there a change a "change in style" or a "change in court membership?"

**Reasons To Accept:**

- The data here is very interesting.  The annotated data is a style of "reasoning" (in contrast to, say, sentiment).  This is a nice setting to test LLMs.
- The comparison across models is comprehensive (i.e., not just a variety of models, but a variety in how the task is structured). I found this insightful and the conclusions compelling.
- The quantitative social science component that quantifies the style shift in the court is both well done and interesting.

**Reasons To Reject:**

- The models do not end up performing particularly well

**Reproducibility:**

4: Could mostly reproduce the results, but there may be some variation because of sample variance or minor variations in their interpretation of the protocol or method.

**Reviewer Confidence:**

4: Quite sure. I tried to check the important points carefully. It's unlikely, though conceivable, that I missed something that should affect my ratings.

---

> ### Author Rebuttal · Authors · 2023-08-29
>
> Thank you for the thoughtful review!
>
> *Concern: Is there any interesting time series in the cross-section? For example, perhaps is a plot of variance (st.dev) across time.*
>
> We agree there are many substantive questions to pursue with the legal reasoning scores developed in this paper. Another working paper of ours looks at the “purity” in style over time, which is closely related to your point about the standard deviation. We find that the transitional periods tend to be less pure, as jurists adapt to new modes of thought.
>
> *Concern: In the time-series plot in Figure 3, is there a "change in style" or a "change in court membership?"*
>
> This is another great question. We also have on our radar, but have not fully explored, the question of whether changes in average style reflect primarily changes in the composition of the court, or instead changes in style holding the court composition fixed. Our initial results suggest it is a mix of the two processes. For instance, the jump in style in 1937 corresponds to a large-scale change in court composition, with a host of new, more liberal justices. But there also appear to be changes in style even accounting for composition.

---

### Meta-Review · Area_Chair_vkGy · 2023-09-17

**Recommendation:** 3

**Metareview:**

This paper makes an important contribution by rigorously evaluating the performance of language models on a complex legal reasoning classification task. The key technical innovation is a high-quality dataset annotated by legal experts categorizing passages of US Supreme Court opinions as either formal/textualist or grand/purposivist interpretive reasoning. This granular labelling required extensive domain knowledge and deliberation, even for human annotators.

The core experiments compare several model architectures, including BERT, DistilBERT, GPT-3, and FLAN-T5. The striking finding is that most models struggle to match human performance levels absent fine-tuning on this in-domain labelled data. Even a strong BERT model fine-tuned on the annotations reaches only ~80% accuracy, indicating inherent challenges in this reasoning style classification task.

In general, reviewers praise the novelty of the dataset and approach, but identify areas for improvement:

Additional SOTA models could be tested to comprehensively benchmark performance. The authors sufficiently address this by including results for models like LLaMA.
Connections could be drawn to the literal vs. purposive interpretation distinction used in some legal frameworks. Of course, I note that the authors already indicate they will incorporate such parallels to situate the conceptual framing.
Lastly, further analysis into changes in court membership vs. judicial philosophy over time was suggested.

I agree with the reviewers that the key contribution here is the introduction of the dataset. However, the task appears to me to be quite trivial - a classification task with 3 labels! Tasks like these are already well solved even in the legal AI field. Moreover, I find it surprising that a finetuned version of a BERT variant, i.e., LEGAL-BERT, even though confers some domain specificity will be able to match the performance of recent LLMs. I am not convinced and I think the authors need to do more exploratory studies with proper prompting techniques - be it in-context or few/zero-shot learning employing newer/larger models.

---

### Decision · Program_Chairs · 2023-10-07

**Decision:**

Accept-Main

**Comment:**

This paper makes an important contribution by rigorously evaluating the performance of language models on a complex legal reasoning classification task. The key technical innovation is a high-quality dataset annotated by legal experts categorizing passages of US Supreme Court opinions as either formal/textualist or grand/purposivist interpretive reasoning. This granular labelling required extensive domain knowledge and deliberation, even for human annotators.

The core experiments compare several model architectures, including BERT, DistilBERT, GPT-3, and FLAN-T5. The striking finding is that most models struggle to match human performance levels absent fine-tuning on this in-domain labelled data. Even a strong BERT model fine-tuned on the annotations reaches only ~80% accuracy, indicating inherent challenges in this reasoning style classification task.

In general, reviewers praise the novelty of the dataset and approach, but identify areas for improvement:

Additional SOTA models could be tested to comprehensively benchmark performance. The authors sufficiently address this by including results for models like LLaMA.
Connections could be drawn to the literal vs. purposive interpretation distinction used in some legal frameworks. Of course, I note that the authors already indicate they will incorporate such parallels to situate the conceptual framing.
Lastly, further analysis into changes in court membership vs. judicial philosophy over time was suggested.

I agree with the reviewers that the key contribution here is the introduction of the dataset. However, the task appears to me to be quite trivial - a classification task with 3 labels! Tasks like these are already well solved even in the legal AI field. Moreover, I find it surprising that a finetuned version of a BERT variant, i.e., LEGAL-BERT, even though confers some domain specificity will be able to match the performance of recent LLMs. I am not convinced and I think the authors need to do more exploratory studies with proper prompting techniques - be it in-context or few/zero-shot learning employing newer/larger models.